# Suppressive Effects of *Lactobacillus* on Depression through Regulating the Gut Microbiota and Metabolites in C57BL/6J Mice Induced by Ampicillin

**DOI:** 10.3390/biomedicines11041068

**Published:** 2023-04-01

**Authors:** Wan-Hua Tsai, Wen-Ling Yeh, Chia-Hsuan Chou, Chia-Lin Wu, Chih-Ho Lai, Yao-Tsung Yeh, Chorng-An Liao, Chih-Chung Wu

**Affiliations:** 1Research and Development Department, GenMont Biotech Incorporation, Tainan 74144, Taiwan; 2Department of Biochemistry, College of Medicine, Chang Gung University, Taoyuan 33302, Taiwan; 3Department of Microbiology and Immunology, College of Medicine, Chang Gung University, Taoyuan 33302, Taiwan; 4Department of Nursing, Asia University, Taichung 413305, Taiwan; 5Department of Medical Research, School of Medicine, China Medical University and Hospital, Taichung 404333, Taiwan; 6Aging and Diseases Prevention Research Center, Fooyin University, Kaohsiung 83102, Taiwan; 7Department of Food and Nutrition, Providence University, Taichung 43301, Taiwan

**Keywords:** *Lactobacillus*, gut microbiota, depression-related metabolites, C57BL/6J mice

## Abstract

Depression is a medical and social problem. Multiple metabolites and neuroinflammation regulate it. Modifying the gut microbiota with probiotics to reduce depression through the gut-brain axis is a potential treatment strategy. In this study, three anti-depressive potentials of *Lactobacillus* spp. (LAB), including *L. rhamnosus* GMNL-74, *L. acidophilus* GMNL-185 and *L. plantarum* GMNL-141, which combined to produce low dosage LAB (1.6 × 10^8^ CFU/mouse, LABL) and high dosage LAB (4.8 × 10^8^ CFU/mouse, LABH), were administered to C57BL/6 mice induced depression by ampicillin (Amp). A behavioral test of depression, 16S ribosomal RNA gene amplicon sequencing, bioinformatic analysis, and short-chain fatty acid (SCFA) content measurement were executed to investigate the gut microbiota composition, activation of nutrient metabolism pathways, levels of inflammatory factors, gut-derived 5-HT biosynthesis genes, and SCFA levels in C57BL/6 mice. Results showed that after mice were induced by Amp, both LAB groups recovered from depressive behaviors, decreased the abundance of *Firmicutes*, and increased the abundance of *Actinobacteria* and *Bacteroidetes* in the mouse ileum. The prediction of metabolism pathways of microbes revealed the activation of arginine and proline metabolism, cyanoamino acid metabolism, and nicotinate and nicotinamide metabolism were increased, and fatty acid synthesis was decreased in both LAB groups. The LABH groups showed increased levels of acetic acid, propanoic acid, and iso-butyric acid and decreased butyric acid levels in the cecum. LABH treatment increased claudin-5 and reduced IL-6 mRNA expression. Both LAB groups also reduced monoamine oxidase, and the LABH group increased vascular endothelial growth factor mRNA expression. These results showed that the composite of three LAB exerts antidepressant effects by regulating the gut microbiota and modifying the levels of depression-related metabolites in C57BL/6J Amp-treated mice.

## 1. Introduction

There are almost 100 trillion bacteria living in the human intestine [1]. Previous studies showed that gut microbiota has many physiological effects that include improvement in the treatment of different CVDs [2], regulate body weight change [3], and modulate the immune response [4], especially in intestinal inflammation [5]. The diverse gut microbiota predominantly comprises bacteria from three important phyla: *Firmicutes*, *Bacteroidetes*, and *Actinobacteria* [6]. These bacteria influence digestion and absorption in the intestine and regulate cognition and the immune system [7]. The gut microbiota also regulates the function of the central nervous system and has gradually attracted attention. Foster et al. [8] showed that gut microbiota plays an essential role in the gut-brain axis. These gut microbes may influence the physiologic function of the brain and nervous system by producing neurotransmitters and their precursors, secreting and regulating signaling proteins and metabolites, and influencing gut hormone release and brain-derived neurotrophic factors [8,9]. However, when microbiota dysbiosis occurs, these pathways are dysregulated, which is associated with altered blood-brain barrier permeability and neuroinflammation. Neuroinflammation is one of the leading causes of neurological diseases [10]. In addition to influencing chronic inflammation, the gut microbiota induces the release of cytokines from intestinal epithelial cells to promote neuroinflammation [11]. Among the involved pathways, the inflammasome pathway has been linked to diseases characterized by neuroinflammatory conditions, such as multiple sclerosis, Alzheimer’s disease, and Parkinson’s disease, as well as anxiety and depressive-like disorders. Previous studies showed that *Lactobacillus* could improve features of Alzheimer’s disease [12] and Parkinson’s disease [13] but also anxiety and depressive-like disorders [14]. Similarly, chronic probiotic treatment substantially improves the depressive phenotype induced by maternal separation in adult offspring [15]. The beneficial effects of probiotic administration are not only observed during conditions of dysbiosis; chronic treatment with a *Lactobacillus* strain reduces anxiety-like and depressive-like behavior during homeostasis and reduces corticosterone reactivity in response to acute stress [16].

Depression is an illness that occurs globally; approximately 280 million people, or 3.8%, have depression. A total of 5.0% of adults and 5.7% of adults older than 60 [17] have depression. Suicide, which is driven by depression, is the fourth leading cause of death in 15- to 29-year-olds [18]. Depression is not only a medical concern but also a social problem. Previous studies have shown that the composition of the microbiota in the host gut might promote or alleviate the pathogenesis of depression [19,20]. Moreover, probiotics could counteract gut dysbiosis and produce anti-depressant-like effects [21]. Among different probiotics, *Lacticaseibacillus* can alleviate *Escherichia coli*-induced depression and reduce cognitive impairment in mice by regulating IL-6 expression and the gut microbiota [22]. *Lactobacillus* in the gut has been shown to prevent postpartum depression via the microbiota-gut-brain axis in a rat model [23]. *Lactobacillus* also improves the depression-like symptoms induced by dendritic cytokine deficiency [24]. Therefore, screening and assessing potential *Lactobacillus* species may contribute to the prevention and treatment of depression. Furthermore, the biochemical metabolism of fatty acids [25], nutrients [26], and phytochemicals [27] have all been shown to regulate the course of depression.

The aim of this study was to screen and assess potential *Lactobacillus* species for anti-depressant effects and analyze the effect of these *Lactobacillus* species on gut microbiota profiles, nutrient metabolism pathways, and the levels of short-chain fatty acids, inflammatory factors, and gut-derived 5-hydroxytryptamine (5-HT) biosynthesis genes.

## 2. Materials and Methods

### 2.1. Lactobacillus Strains and Culture

As shown in Table 1, 10 individual *Lactobacillus* strains were obtained from the Probiotic Bank (GenMont Biotech, Tainan, Taiwan). The lyophilized live probiotic powder was resuspended in sterile Milli-Q water, mixed well by vortexing, and then administered orally.

### 2.2. Animal Feed

Seven-week-old male C57BL/6J mice (*n* = 108) were purchased from the National Laboratory Animal Centre (Taipei, Taiwan) for experimental models established in two stages. The study protocol was approved by the IACUC Laboratory Animal Center of GenMont Biotech Incorporation (Taiwan IACUC Approval No. 194 GenMont Biotech Incorporation Approval Nos. 108006, 108009, 109003, 109006, and 110005). All the experiments were performed in accordance with the Care and Use of Laboratory Animals (8th Ed. National Research Council (US) Committee 2011) and the Institutional Animal Care and Use Committee (IACUC) of GenMont Biotech Incorporation (Tainan, Taiwan).

### 2.3. Animal Treatment-Prescreening and Identification of the Lactobacillus Strain with the Most Potential for Recovering Depressive Behavior

The first stage of experiments performed to screen LAB species with high therapeutic potential involved behavioral tests to assess depression. The model of depression and the test used to evaluate depressive-like behavior were established using modifications of the induce and assessment methods described by Jang et al. [28] and Ceylani et al. [29], respectively. After 1 week of acclimation, the mice (*n* = 80) were randomly assigned to 12 groups by body weight. As shown in Figure 1, the normal control (NC) group (*n* = 10) and Amp group (Amp) (*n* = 10) mice were orally administered phosphate-buffered saline (PBS) or 100 mg/kg/day Amp, respectively, once per day from the 1st day until the 2nd day. After Amp treatment, mice were orally administered an equal volume of Milli-Q water (NC and Amp groups) or 10 LAB strains from the 3rd day until the 7th day (each group *n* = 6). All groups were assessed for depressive behavioral changes on the initial (0), 8th and 12th days.

According to the recovery from depression-like behavior rate evaluated during the first stage test (Table 1), three of the strains associated with the best recovery rates, including A1, A4, and A5, which possessed the most potent activities, were identified, and the species associated with each strain were evaluated. The A1 and A5 species included *L. rhamnosus* GMNL-74 (BCRC 910236; CCTCC M 203098) and *L. acidophilus* GMNL-185 (BCRC 910774; CCTCC M 2017764), respectively, as determined by 16S rDNA sequencing and API50 analysis [30]. The A4 species was *L. plantarum* GMNL-141 (BCRC 911066; NITE BP-03510), which was determined by 16S rDNA sequencing and API50 analysis (Appendix A).

As shown in Table 1, after the behavior recovery rate was assessed during the first stage test, the seven-week-old male C57BL/6J mice (*n* = 36) were randomly assigned to 4 groups by body weight. In the NC group (*n* = 9) and Amp group (*n* = 10), mice were orally administered PBS or 100 mg/kg/day Amp, respectively, once per day from the 1st day until the 2nd day. After Amp treatment, mice were orally administered an equal volume of Milli-Q water (NC and Amp groups) or a strain composite (A1, A4, and A5) and were fed a dose of 1.6 × 10^8^ CFU/mouse (low dosage, LABL) (*n* = 12) or 4.8 × 10^8^ CFU/mouse (high dosage, LABH) (*n* = 12) from the 3rd day until the 7th day as experimental groups. Four groups were assessed for depressive behavioral changes on the initial (0), 8th and 12th days. Then, all mice were sacrificed with carbon dioxide on the 12th day. The ileum, cecum, and descending colon were removed, weighed, and inspected prior to pathological, biochemical, and gut microbiota analysis.

### 2.4. Behavioral Test for Depression

#### 2.4.1. Nest-Building Test

In this study, the assessment of the shredding procedure conducted during the nest-building test, which is one of the behavioral tests used to evaluate depression, was performed as previously described [29]. C57BL/6J mice were acclimated to the plastic experimental cage for 1 h under normal overhead fluorescent lighting. Then, mice were transferred into another plastic experimental cage with 5 mm sawdust shavings on the floor and a preweighed multiply gauze nestled (Ancare, Bellmore, NY, USA). The nestlet comprised 5 cm × 5 cm, 5 mm thick cotton fibers and weighed approximately 3 g. Mice were left in the cage for 60 min, after which the weight of the gauze pad that the mouse had not removed was measured. The weight of the gauze used for nest construction was determined by subtraction.

#### 2.4.2. Forced Swimming Test (FST)

The aim of the FST was to evaluate the extent of despair in mice according to the methods described by Bergner et al. [31]. The mice swam in a beaker (20 × 14 cm in diameter) containing 16 cm deep water at 24 ± 1 °C. The cumulative immobility time (s) was recorded for the last 4 min of a 6-min swim test. If a mouse stopped struggling or produced only small limb movements to keep its head floating above the water, it was considered motionless.

### 2.5. 16S Ribosomal RNA Gene Amplicon Sequencing

In this study, 16S ribosomal RNA gene amplicon sequencing was performed as previously described [32]. Ileum samples were collected from each group after 12 days, and all specimens were extracted using a Qiagen DNA kit (Qiagen, Hilden, Germany) according to the manufacturer’s instructions. In addition, to precisely evaluate the data used for bioinformatics analysis and exclude abnormal data, we collected taxonomic composition data from 6 mice in each group.

A fragment analyzer (5300; Agilent Technologies, Santa Clara, CA, USA) was used for the size of the amplified DNA product, and sequencing was conducted using an Illumina MiSeq platform in this study. The 16S ribosomal RNA (rRNA) PCR was performed using metagenomic DNA as a template, which was amplified with the bacteria-specific primers S17 (5′-TCG TCG GCA GCG TCA GAT GTG TAT AAG AGA CAG CCT ACG GGN GGC WGC AG-3′) and A21 (5′-GTC TCG TGG GCT CGG AGA TGT GTA TAA GAG ACA GGA CTA CHV GGG TAT CTA ATC C-3′).

The Nextera XT Index Kit v2 was used to assign indices and Illumina sequencing adapters of DNA samples. The samples were mixed using a 600-cycle MiSeq Reagent Kit v3 at a final concentration of 4 pM after library construction. Then loaded onto a MiSeq cartridge and transferred onto the instrument. An automated cluster generation and a 2 × 300 bp paired-end sequencing run were performed.

The qualified reads were obtained from the sequences generated after passing a filtering process, and operational taxonomic units (OTUs) at a 97% similarity with the Greengenes database (v13.8) were clustered. And a Qiagen CLC Microbial Genomics Module (v10.1.1) for further analysis.

### 2.6. Bioinformatic Analysis

The processing and statistical analysis of meta-taxonomic data were performed as previously described [32]. To determine the difference in microbial composition among groups, the Shannon diversity index was used to analyze the alpha diversity of taxonomic composition. The principal Coordinate Analysis (PCoA)-weighted UniFrac was used to measure the Beta diversity.

The Qiagen CLC Microbial Genomics Module and linear discriminant analysis effect size (LEfSe) were combined to produce the OTU table. To identify specific microbial markers among groups, tLEfSe was performed using this study’s Galaxy/Hutlab web tool. A 0.05 alpha value as the cutoff for the factorial Kruskal–Wallis test, pairwise Wilcoxon test, and a linear discriminant analysis (LDA) score cutoff of 2.0. The abundance across each sample was used to reconstruct the metagenome and infer the microbial functions through PICRUSt2 (version 2.3.0-b) [33]. Principal component analysis (PCA; performed using the ade4 package v1.7-16) was applied using the R language (v4.0.2).

### 2.7. Measurement of Short-Chain Fatty Acid (SCFA) Concentration in Mouse Cecal Contents

SCFA analysis was carried out using gas chromatography using derivatized samples as previously described in Niccolai et al. [34]. A 0.2–0.4 g sample of the cecum content (*n* = 6/group) was dissolved in 10 mL reverse osmosis (RO) water and vortexed for 2 min. Then, the mixture was centrifuged at 120× *g* for 15 min. The suspension was collected and filtered through a 0.45 μm filter. Two milliliters of the filtered sample were acidified by 2.0 mL 50% sulfuric acid, extracted by 2 mL ether by a vortex device for 2 min, and centrifuged at 120× *g* for 5 min. After sitting at 4 °C for 30 min, the suspension (ether fraction) was used for gas chromatography–mass spectrometry (GC/MS) analysis. 

As previously described in Niccolai et al. [34], the analysis was accomplished with Technologies 6850 GC with a split/splitless injector and flame ionization detector (FID). A GC 6890N equipped with a 5973 mass spectrometer detector (Agilent Technologies, Santa Clara, CA, USA) was also used to confirm the identity of the analytes in the samples. The capillary chromatographic column used was a nitroterephthalic acid-modified polyethylene glycol (PEG) column (DBFFAP, 30 m, 0.25 mm i.d., 0.25 µm film thickness, purchased from Agilent Technologies). Each SCFA standard compound was used for quantitative analysis. The levels of each SCFA were measured using units of “μmole/g cecum content.” Then, the SFCA levels were presented as “the percentage of individual SCFA/total SCFA levels”.

### 2.8. Analysis of RNA Expression of Intestinal Tight Junction Protein-Related, Inflammatory Cytokine, and Gut-Derived 5-HT Biosynthesis Genes

To specifically measure the expression of intestinal tight junction protein-related genes, inflammatory cytokine genes, and gut-derived 5-HT biosynthesis genes, we collected data from 4–5 mice in each group. The front section of the ileum samples obtained from the experimental mice was used for RNA extraction [32]. After washing with sterilized cold PBS to remove stool particles, the mucosa and intestinal wall of the ileum were collected by a scraper and stored at −80 °C and used as samples for RNA isolation. After the purified RNA was quantitative, the RNA was stored at −80 °C until RNA expression analysis. A total RNA sample (5 μg) was used for cDNA synthesized in this study.

Quantitative real-time reverse transcription-polymerase chain reaction (PCR) was used to analyze the levels of intestinal tight junction protein-related genes, inflammatory cytokines, and gut-derived 5-HT biosynthesis genes. As shown in Table 2, the intestinal tight junction protein-related genes, inflammatory cytokine genes, and gut-derived 5-HT biosynthesis gene, including Claudin2, Claudin3, Claudin5, IL-6, TNF-α, IFN-γ, Tph1, Mao, Pten, Vegfα, and glyceraldehyde-3-phosphate dehydrogenase (GAPDH), were expressed, and quantitated by using 2× Rotor-Gene SYBR Green PCR Master Mix (204076; Qiagen, Hilden, Germany). A final reaction mixture (10 μL) was prepared from mixing aliquots of cDNA, 3 μL of forward and reverse primers, and 5 μL of 2× Rotor-Gene SYBR Green PCR Master Mix.

Real-time PCR data for the levels of each gut tight junction protein, inflammatory cytokines, and gut-derived 5-HT biosynthesis genes were calculated by relative calculation using the delta Ct method (2-ΔΔCt). ΔCt was calculated as the difference between the Ct value measured using the primers for a specific target gene and the Ct value measured using the primers for GAPDH. ΔΔCt was defined as the difference between the ΔCt value measured in each treatment and the ΔCt value measured in the NC group. The values derived from the 2-ΔΔCt method show the fold changes between a treated sample and the NC group.

### 2.9. Statistical Analysis

In this study, the present data were calculated and visualized using GraphPad Prism 8.0.2 software (San Diego, CA, USA). Statistically significant differences in the behavioral test results were evaluated using a one-way analysis of variance (ANOVA) and Dunnett’s post hoc test for multiple comparisons. The Kruskal-Wallis test with Bonferroni correction was performed for gut microbial data analysis to determine significant changes in microbiota species. The Wilcoxon test was used to evaluate differences in the α-diversity index among groups, and the β-diversity index results were analyzed by nonparametric analysis of similarities (ANOSIM). Finally, the Student’s *t*-test was used to calculate the statistical significance of the differences in the levels of SCFAs, intestinal tight junction protein-related and gut-derived 5-HT biosynthesis genes, and inflammatory cytokine genes between the two groups.

## 3. Results

### 3.1. Composite Lactobacillus spp. Improves Features of Amp-Induced Depression in C57BL/6J Mice

After the first stage of depression behavior recovery, mice treated with the A1, A4, and A5 strains had higher depression recovery rates than mice treated with the other strains (Table 1). After 16S rDNA sequencing and API50 analysis, the A1, A4, and A5 strains were identified to be *L. rhamnosus* GMNL-74, *L. plantarum* GMNL-141, and *L. acidophilus* GMNL-185, respectively. Therefore, the following experiments involved the combined use of these three LABs with high anti-depressant potential to establish composite low-dosage and high-dosage LAB samples.

As shown in Figure 2a, the nestlet shred weight was significantly lower in the group treated with Amp for eight days than in the NC group (*p* < 0.05). This result shows that Amp treatment successfully induced depression in mice. In addition, the nestlet shred weight levels in the LABH group were significantly higher than those in the Amp group (*p* < 0.05) after eight days of treatment. After 12 days of treatment, both the LABL and LABH groups had a significantly higher nestlet shred weight than the Amp group (*p* < 0.05). In addition, during the forced swimming test (escape behavior), both the LABL and LABH groups showed significantly higher escape time than the Amp group (*p* < 0.05) after 8 and 12 days of treatment (Figure 2b). These data show that these three groups administered LAB composite treatment had a higher rate of recovery from depressive behavior than those treated with individual strains (Table 1). The improvement of depressive behavior mediated by LABH was more pronounced than that mediated by LABL.

### 3.2. Composite Lactobacillus spp. Regulates Gut Microbiota Composition in C57BL/6J Mice Treated with Amp

The intestinal samples obtained from 6 mice in 4 groups were used for 16S rRNA analysis of intestinal bacteria. According to the OTU heatmap, the changes in the phyla OTUs (Figure 3a) represented the average change in the intestinal flora of C57BL/6J mice. In the Amp group, the abundance of *Firmicutes* and *Verrucomicrobia* increased, but the abundance of Actinobacteria significantly decreased (*p* < 0.05) (Figure 3b–d). In contrast, in the LABL and LABH groups, there was a decrease in *Firmicutes* abundance, but *Bacteroidetes* abundance was significantly increased (*p* < 0.05) (Figure 3b,e). The *Actinobacteria* abundance was significantly higher in the LABH group than in the Amp group (*p* < 0.05) (Figure 3d). In addition, the *Bacteroidetes*/*Firmicutes* (B/F) ratio was significantly higher in the LABL and LABH groups than in the Amp group (*p* < 0.05) (Figure 3f).

Regarding alpha diversity, there were no significant differences in species diversity between the groups (Figure 3g). The beta diversity differences between the NC and Amp, NC and LABL, and NC and LABH groups were significant (*p* < 0.05), indicating that there were composition differences between the groups (Figure 3h). The beta diversity analysis also showed that species composition in the Amp and LABL and the LABL and LABH groups were significantly different (*p* < 0.05). This result indicated that LAB could change the gut microbiota and that these changes differ from those observed following Amp treatment. To evaluate the association between the gut microbiota and depression, we performed a taxonomic analysis of the gut microbiota in these four groups.

A LEfSe analysis-based cladogram was generated for the gut microbiota (Figure 4a), and PCA between dominant bacterial genera from gut microbiota was performed (Figure 4b), and Figure 4c showed an LDA score (4c). The average changes in the order, family, and genus OTUs in the intestinal flora of mice were analyzed. At the level of taxonomic order, the Amp group had significantly higher *Bacillales* abundance than the NC group, but the *Bacillales* abundance in the LABL and LABH groups was significantly lower than that in the Amp group (*p* < 0.05) (Figure 4d), and the abundance in these groups was not significantly different from that in the NC group. At the family level, Amp treatment significantly increased *Verrucomicrobiaceae* abundance (*p* < 0.05) (Figure 4e). Notably, the *Enterobacteriaceae* levels increased, and the *Lactobacillaceae* levels decreased after Amp treatment (Figure 4f,g). However, the LABH group showed higher *Lactobacillaceae* abundance than the Amp group (*p* < 0.05) (Figure 4g), and the LABL and LABH groups showed lower *Enterobacteriaceae* abundance than the Amp group (*p* < 0.05) (Figure 4f). *Erysipelotrichaceae* abundance in LABL and LABH groups was significantly higher than that in the NC group (*p* < 0.05) (Figure 4h), but the levels were not significantly different from levels in the Amp group. As shown in Figure 4i, at the genus level, *Prevotella* abundance in the LABL and LABH groups was significantly lower than that in the NC group (*p* < 0.05), but the levels were not significantly different from levels in the Amp group.

The Amp group had significantly higher *Staphylococcus* abundance and lower *Ruminococcus* abundance than the NC groups (*p* < 0.05) (Figure 4j,k). In contrast, the *Staphylococcus* abundance in the LABL and LABH groups was significantly lower than that in the Amp group (*p* < 0.05), and the *Ruminococcus* abundance was significantly higher in the LABH group than in the Amp group (*p* < 0.05) (Figure 4j,k). In addition, the LABH group had a significantly higher abundance of *Ruminococcus*, *Bifidobacterium*, *Lactobacillus*, *Dorea,* and *Desulfovibrio* than the Amp group (*p* < 0.05) (Figure 4k–o). At the species level, Amp treatment significantly decreased *Lactobacillus* (unidentified) abundance (*p* < 0.05) (Figure 4p). In addition, the abundance of *Lactobacillus* (unidentified) and *Desulfovibrio* C21-c20 was significantly higher in the LABH group than in the Amp group (*p* < 0.05) (Figure 4p,r). Both the LABL and LABH groups showed significantly higher *S24-7* abundance than the Amp group (*p* < 0.05) (Figure 4r).

### 3.3. Composite Lactobacillus spp. Regulates Gut Microbe-Related Pathways in C57BL/6J Mice Treated with Amp

PICRUSt2 was used to analyze further the regulatory pathways in which the gut microbes were involved, and STAMP (v2.1.3) was used to map the resulting signal transmission pathways in this study. The Venn diagram showed that there were 102 pathways affected by the gut microbiota in each group (Figure 5a).

The Amp group was compared with the LABL group, and there were 100 pathways affected by the gut microbiota that differed between the LABL group and the Amp group. Approximately 81 pathways differed between the Amp group and the LABH group (Figure 5a). The pathways intersected, and 49 of them were pathways with common changes (Table 3); thus, 49 detailed pathways and ratios were compared. The results presented an increasing (I) or a decreasing (D) trend, as shown in Table 3. Some of the pathways involved in improving anxiety attracted our attention. Both the LABL and LABH groups had significantly higher expression of genes involved in arginine and proline metabolism, cyanoamino acid metabolism, flavone, and flavonol biosynthesis, isoquinoline alkaloid biosynthesis, and nicotinamide metabolism than the Amp group (*p* < 0.05) (Figure 5b–f). Both the LABL and LABH groups had significantly lower expression of genes involved in fatty acid biosynthesis, linoleic acid metabolism, and the *Staphylococcus* aureus infection pathway than the Amp group (*p* < 0.05) (Figure 5g–i).

### 3.4. Composite Lactobacillus spp. Regulates Fecal Fatty Acid Composition in C57BL/6J Mice Treated with Amp

Figure 6 shows the percentages of SCFA in the cecum of C57BL/6J mice. The results showed that the most important fatty acid was acetic acid, and the joint second most important were the butyric and propanoic acids; the SCFA with the lowest amount was isobutyric acid in the C57BL/6J mice (Figure 6). In the Amp group, the propanoic acid (PA) level was significantly higher, and butyric acid (BA) was significantly lower than in the NC groups (Figure 6a,b) (*p* < 0.05). The LABL groups also had significantly higher levels of acetic acid (AA), propanoic acid, and iso-butyric acid (Iso-BA) and significantly lower levels of butyric acid than the NC groups (*p* < 0.05) (Figure 6). Acetic acid levels were significantly higher, and butyric acid levels were significantly lower in the LABL group than in the Amp groups (Figure 6b,c) (*p* < 0.05). Additionally, the LABH groups had significantly higher levels of acetic acid, propanoic acid, and isobutyric acid and significantly lower levels of butyric acid than both the NC and Amp groups (Figure 6) (*p* < 0.05).

### 3.5. Composite Lactobacillus spp. Regulates Intestinal Tight Junction, Inflammatory Response Factors, and Gut-Derived 5-HT Biosynthesis Genes in C57BL/6J Mice Treated with Amp

The mRNA expression of intestinal tight junction protein genes and inflammatory response factors is shown in Figure 7. The Claudin2 and Claudin3 mRNA expression levels were not different among the NC, Amp, LABL, and LABH groups (Figure 7a,b). The mRNA expression of Claudin5 was significantly higher in the LABL and LABH groups than in the Amp group (*p* < 0.05) (Figure 7c).

Regarding the mRNA levels of inflammatory response factors, the LABH group had significantly lower mRNA expression of IL-6 than the Amp group (*p* < 0.05) (Figure 7d). However, there was no difference in TNF-α and INF-γ expression between the NC, Amp, LABL, and LABH groups (Figure 7e,f). In addition, both LAB groups were significantly lower mRNA expression of Mao than the Amp (*p* < 0.05) (Figure 7h). The LABL group had a significantly lower mRNA expression of Pten than the Amp group (*p* < 0.05) (Figure 7i). The LABH group had a significantly higher mRNA of Vegfα than the Amp group (*p* < 0.05) (Figure 7j).

## 4. Discussion

The results of the present study demonstrated that LABL and LABH treatment have anti-depressive potential and that LABH treatment is more efficient in alleviating depression. The mechanism may involve the regulation of the composition of the gut microbiota, modification of the mRNA expression of genes involved in depression-related metabolic pathways, regulation of the levels of SCFAs, reduction of the levels of inflammatory factors mediated and gut-derived 5-HT biosynthesis by the identified *Lactobacillus* spp. In addition, most studies on the relationship between gut microbiota and neurological disease have usually focused on a single bacterial species. The use of three *Lactobacillus* spp., including *L. rhamnosus* GMNL-74, *L. plantarum* GMNL-141, and *L. acidophilus* GMNL-185, in this study, represents a novel approach that a composite microbial formula for investigating the potential anti-depressive effects of *Lactobacillus* spp.

It is known that there is a strong connection between chronic stress, microbiota changes, activation of the inflammatory response, and depression [35]. Previous studies showed that gut microbiota composition was significantly altered, particularly in the composition of *Firmicutes*, *Actinobacteria*, and *Bacteroidetes* at the phylum level in patients with major depressive disorder (MDD) and in mouse models [36]. In the social defeat stress (SDS) and chronic unpredictable mild stress (CUMS) mice [37,38,39] and clinic MDD patients [40,41] all showed that the substantially high correlation between the depression-like behaviors and gut microbiota composition, including from intestine and/or fecal samples. Their studies have shown an increased *Firmicutes*, *Sutterella*, *Oscillospira*, *Bacteroidetes*, *Proteobacteria*, *Actinobacteria*, and *Verrucomicrobia* abundance in intestines and/or fecal under depression-like behaviors. Notably, both LABL and LABH treatment reduced the abundance of *Firmicutes* and *Verrucomicrobia*, with positive correlations with depression in this study. LABL and LABH treatments can recover *Actinobacteria* and *Bacteroidetes* abundance, which both have negative correlations with depression and increase the B/F ratio, which is an indicator of factors with negative correlations with depression. In addition, the gut microbiota of C57BL/6 mice treated with Amp had an increased abundance of pathogens, including *Bacillales* [38], *Verrucomicrobiaceae* [39], *Staphylococcus* [42], and *Enterobacteriaceae* [43]. However, in this study, both LABL and LABH reduced the abundance of these gut microbes. Additionally, the gut microbiota of C57BL/6J mice treated with Amp had a reduced abundance of probiotics, including *Lactobacillaceae* [40], *Lactobacillus* [44], *Bifidobacterium* [45], *Ruminococcus* [46], and *S24-7* [47]. LABH treatment increased the abundance of these gut microbes. This treatment has the potential to promote recovery from depression induced by Amp.

However, many studies have shown that dietary probiotics and supplements can regulate the gut microbiota and attenuate anxiety and depression syndrome. Ramalho et al. [48] used *Lactococcus lactis* subsp. *cremoris* LL95 to ameliorate mood disorders in a lipopolysaccharide (LPS)-induced depression-like C57BL/6J mouse model. *Lactococcus* can reduce depression syndrome by reducing the levels of reactive oxygen species, tumor necrosis factor-α, and interleukin-1β in the hippocampal tissues of these animals and changing the fecal lactic acid bacteria content. In a postpartum depression rat model, gavage-fed *Lactobacillus casei* improved depressive-like behaviors by altering gut microbiota composition, brain monoamines, and oxidative stress, which may be associated with the regulation of the BDNF-ERK1/2 pathway [49]. The present study and previous studies have shown that probiotics and dietary supplements may regulate the gut microbiota to reduce depression. However, the specific methods for regulating the composition of the gut microbiota and the brain and neurologic function by targeting the gut–brain axis have become a hot issue.

In the present study, both the LABL and LABH groups had significantly increased the expression of genes involved in arginine and proline metabolism, cyanoamino acid metabolism, and nicotinate and nicotinamide metabolism and significantly decreased the expression of genes involved in the fatty acid biosynthesis pathway. These intestinal metabolism pathways are regulated, and the produced metabolites affect cytokine release, neurotransmitter production, and specific amino acid and short-chain fatty acid contents, leading to behavioral changes, such as anxiety and depression, by the gut-brain axis [4,5,7,16,19]. These regulate neuro-and-brain metabolites effects, including regulating the GABA levels in the brain [50], stimulation of serotonergic, dopaminergic, and noradrenergic neurotransmission [51,52], regulating energy metabolism [53], and neuroprotection [54,55]. Notably, this study is the first to show that the pathways of flavonol biosynthesis and isoquinoline alkaloid biosynthesis were inhibited in a mouse model of depression. The LAB groups showed significantly higher activation of flavonol biosynthesis and isoquinoline alkaloid biosynthesis metabolic pathways than the Amp group. Previous studies have shown that dietary flavonol, isoflavone, and isoquinoline alkaloids can alleviate neurological disorders and depression [56,57]. Increasing the activation of flavonol biosynthesis and isoquinoline alkaloid biosynthesis metabolic pathways may underlie the potential anti-depressive effects of LAB. The microbiota interacts with the brain through the gut-brain axis, and the metabolites produced by the gut microbiota regulate the brain’s physiological functions and behaviors. Modification of the above metabolic pathways may be involved in the anti-depressive effects of LABL and LABH.

In addition, serotonin (5-hydroxytryptamine, 5-HT), which is a neurotransmitter, plays an important role in anti-depression [58]. There is 95% of 5-HT is produced in the peripheral system, especially in the gastrointestinal tract [59]. In the gastrointestinal tract, tryptophan hydroxylase (Tph) is predominantly found in EC cells. It is a rate-limiting enzyme for 5-HT and plays a key role in the conversion of tryptophan to 5-hydroxytryptophan (5-HTP) [58]. Then, the monoamine oxidase (MAO) is catalyzed 5-HT into 5-hydroxy indole acetic acid (5-HIAA), which is finally excreted in urine [60]. In this present study, although the Tph1 mRNA levels are non-effect by these two LAB groups. However, the Mao mRNA levels are significantly lower than the Amp induce group in this present study. LAB helps the EC maintain the level of 5-HT in the GI tract and not for further metabolization. It may be a reason that LAB increases 5-HT and helps anti-depression.

Interestingly, vascular endothelial growth factor (VEGF) is a mediator of both the neurogenetic and behavioral action of anti-depressants [61]. The clinical patient has a higher hippocampus and neuronal VEGF after anti-depressant treatment [62]. On the other hand, over-expression of phosphatase and tensin homolog deleted on chromosome ten (Pten) in the prefrontal cortex led to an increase in depression-like behaviors, whereas genetic inactivation of Pten in the prefrontal cortex prevented the depression-like behaviors [62]. In the present study, the LABL group significantly reduced the mRNA levels of Pten but increased Vegfα mRNA expression. These Mao, Pten, and Vegf results show LAB has a high potential to maintain serotonin metabolism for anti-depression.

The gut microbiota regulates not only intestinal metabolic pathways but also changes fecal SCFA production. SCFAs are the main metabolites produced by bacterial fermentation of dietary fiber in the gut microbiota. SCFAs play an important role in microbiota-gut-brain crosstalk [63,64]. Acetic acid, propionic acid, and butyric acid are major high-abundance fatty acids in the human intestine, but formate, valerate, and caproate are produced at relatively low levels [63]. SCFAs can regulate plasma hormone levels and enter the BBB to induce histone crotonylation and the regulation of inflammatory and immune responses by cellular NF-κB and AMPK signaling in the central nervous system, which is involved in multiple psychological and physiological functions [64]. In addition to intestinal AA, PA, and BA, Iso-BA were found in the intestine of animals in the LABH group in the present study. A previous clinical trial showed that the acetic acid content in depressed women was significantly lower than that in nondepressed women, while the concentration of isocaproic acid was significantly higher than that in matched healthy subjects [65]. Wu et al. [66] also demonstrated changes in SCFAs in depressed mice subjected to a chronic restraint stress (CRS) model. Acetic acid, propionic acid, and pentanoic acid levels were significantly lower in depressed mice than in control mice. In the present study, the LABL and LABH groups had significantly higher acetic acid levels than the Amp group, and the LABH group also had higher PA and Iso-BA levels. In addition, we analyzed the relationship between SCFA levels, the gut microbiota, and depression-like (nestlet shredding and escape behaviors) after LAB treatment. Appendix A shows that there is a significantly negative relationship between the PA content and escape behavior (*p* < 0.05). In addition, the *Dorea* abundance (%) and level of nestlet shredding (g) had a significantly negative correlation (*p* < 0.05). These results indicated that LAB could regulate gut SCFA levels and that microbial levels affect recovery from amp-induced depressive behavior. Improved levels of AA, PA, and iso-BA in the ileum may have been involved in the amelioration of depression observed in this study.

The balanced interplay between the epithelial barrier, immune system, and microbiota maintains gut homeostasis; disrupting this interplay may lead to inflammation. Paracellular permeability is governed by tight intercellular junctions. Both gut microbiota composition and SCFAs maintain intestinal barrier integrity and protect against inflammation [67,68]. Miranda-Ribera et al. [69] used a zonulin transgenic mouse (Ztm) model and found that these animals exhibited a profile of tight junction gene expression in the gut that differed from that in wild-type (WT) mice: Claudin-15, Claudin-5, Jam-3, and Myosin-1C levels were decreased in the duodenum of male mice. Supplemental xylooligosaccharide can remodel the cecal microbiota of animals fed the OFO diet, including increases in the abundance of *Firmicutes*, *Ruminococcaceae*, *Verrucomicrobia* (Akkermansia), *Paraprevotella*, *Prevotella_9*, and *Oscillospira*, along with a decrease in the abundance of *Erysipelatoclostridium*. Increased Claudin5 levels in the group treated with xylooligosaccharide combined with XOS promoted and maintained intestinal integrity [70]. In the present study, the levels of claudin-5 mRNA were significantly higher in the LABL and LABH groups than in the Amp group. The LABH treatment significantly reduced IL-6 expression, which prevented damage from the production of proinflammatory factors.

This present study shows a new perspective on the influence of these three combined *Lactobacillus* on depression-like behavior. Further studies can focus on investigating the molecular and gene expression mechanisms of inflammatory factors, cell junction proteins, and neurotransmitters on the gut–brain axis metabolism. According to our findings and the considerations outlined above, clinical trials of lactobacillus application are needed for the optimal dose of these three *Lactobacillus* spp., and the regulation mechanisms study is needed in the future.

## 5. Conclusions

These results demonstrated that composite LAB (*L. rhamnosus* GMNL-74, *L. acidophilus* GMNL-185, and *L. plantarum* GMNL-141) treatment has anti-depressive potential. The potential involvement of LAB in regulating the composition of the gut microbiota, reducing the levels of inflammatory factors, regulating the levels of short-chain fatty acids, modifying the mRNA expression of depression-related metabolic pathways, and gut-derived 5-hydroxytryptamine (5-HT) biosynthesis genes may also be involved.

## Figures and Tables

**Figure 1 biomedicines-11-01068-f001:**
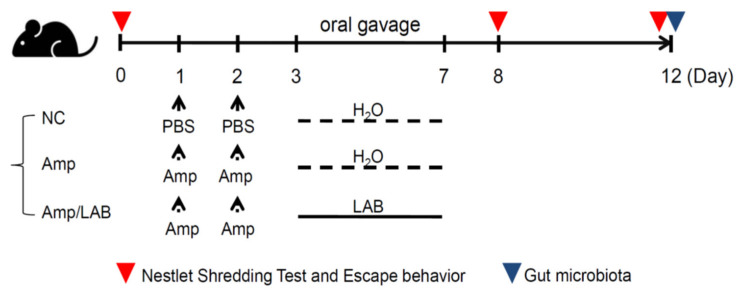
Experimental design of the assessment of the effects of *Lactobacillus* spp. on depression behaviors in C57BL/6 mice.

**Figure 2 biomedicines-11-01068-f002:**
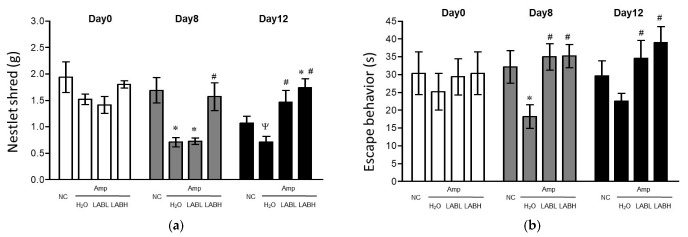
Assessment of the effects of *Lactobacillus* spp. on depression behaviors in C57BL/6 mice. C57BL/6J mice were grouped into the normal control (NC) group (*n* = 9) and Amp group (Amp, *n* = 10). The groups treated with a composite formula that combined three strains were fed a dose of 1.6 × 10^8^ CFU/mouse (low dosage, LABL, *n* = 12) or 4.8 × 10^8^ CFU/mouse (high dosage, LABH, *n* = 12) from the 3rd day until the 7th day. All groups were assessed for depressive behavioral changes on the initial (0), 8th and 12th days. Behavioral tests for depression included the nest-building test and forced swimming test. (**a**) Quantification of Nestlet Shredding. (**b**) Quantification of Escape behavior. *, compared with NC group, *p* < 0.05; #, compared with Amp group, *p* < 0.05, Ψ, compared with LABL group, *p* < 0.05.

**Figure 3 biomedicines-11-01068-f003:**
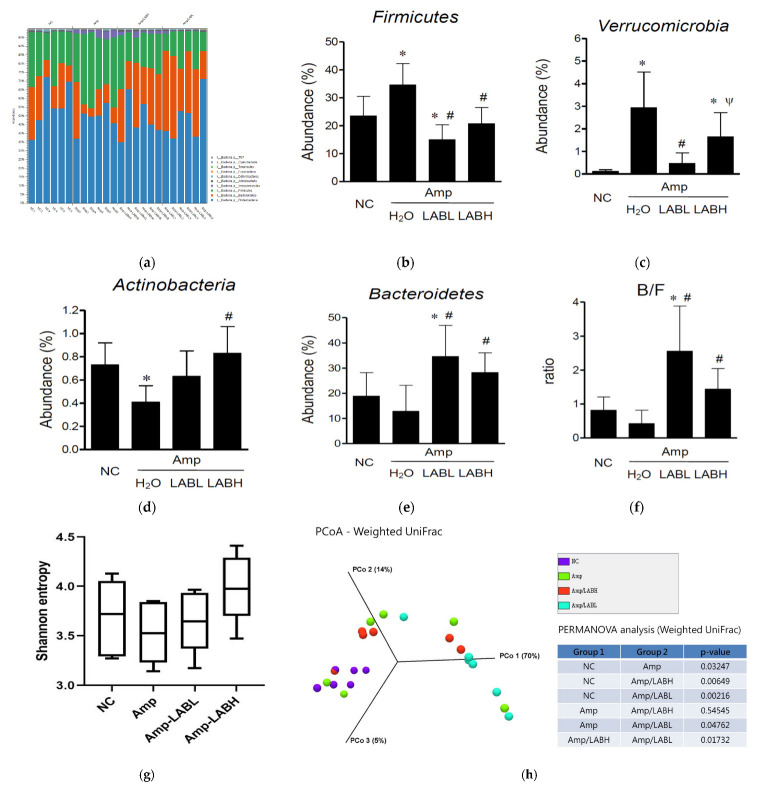
Effect of *Lactobacillus* spp. on gut microbiota composition at the phylum level and diversity and taxonomic profiling in C57BL/6J mice. C57BL/6J mice (*n* = 6/group) were grouped into the NC and Amp groups. The groups given the three-strain combination were fed a dose of 1.6 × 10^8^ CFU/mouse (LABL) or 4.8 × 10^8^ CFU/mouse (LABH) from the 3rd day until the 7th day. All groups were analyzed for 16S ribosomal RNA V3–V4 gene expression in the ileum on the 12th day. (**a**) Phylogenetic cladogram from the LEfSe analysis, (**b**) *Firmicutes* spp., (**c**) *Verruocrobia* spp., (**d**) *Actinobacteria* spp., (**e**) *Bacteroidetes* spp. (**f**) B/F ratio communities in the control and Amp groups. (**g**) α-diversity of intestinal microbial communities. (**h**) β-diversity of intestinal microbial communities. *, compared with NC group, *p* < 0.05; #, compared with Amp group, *p* < 0.05, Ψ, compared with LABL group, *p* < 0.05.

**Figure 4 biomedicines-11-01068-f004:**
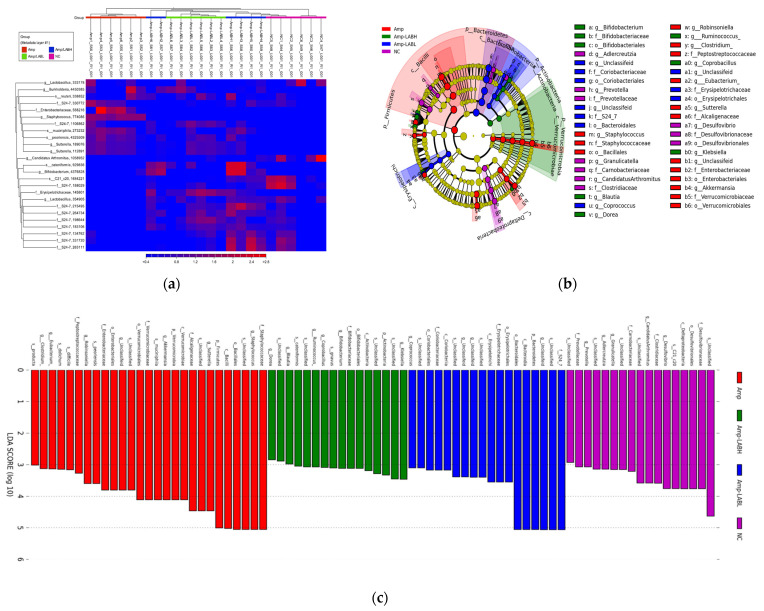
Effect of *Lactobacillus* spp. on gut microbiota composition at the order, family, genus, and species levels in C57BL/6J mice. C57BL/6J mice (*n* = 6/group) were grouped into the normal control (NC) group and Amp group (Amp). The groups administered a composite formula with three strains were fed a dose of 1.6 × 10^8^ CFU/mouse (low dosage, LABL) or 4.8 × 10^8^ CFU/mouse (high dosage, LABH) from the 3rd day until the 7th day. All groups were analyzed for 16S ribosomal RNA V3-V4 gene expression in the ileum on the 12th day. (**a**) OTU heatmap distribution, (**b**) Principal coordinate analysis of Bray–Curtis dissimilarity, (**c**) the microbial community relative LDA score. Order level: (**d**) *Bacillales* spp.; Family levels—(**e**) *Verrucomicrobiaceae* spp., (**f**) Enterobacteriaceae spp., (**g**) *Lactobacillaceae* spp., (**h**) *Erysipelotrichaceae* spp.; genus level: (**i**) *Prevotelia* spp., (**j**) Staphylococcus spp., (**k**) *Ruminococcus* spp., (**l**) *Bifidobacterium* spp., (**m**) *Lactobacillus* spp., (**n**) *Dorae* spp., (**o**) *Desulfovibrio* spp.; species levels: (**p**) *Lactobacillus* (unidentified) spp., (**q**) *Desulfovibrio* C21_c20 spp., (**r**) *S24-7* spp. *, compared with NC group, *p* < 0.05; #, compared with Amp group, *p* < 0.05; Ψ compared with LABL group, *p* < 0.05.

**Figure 5 biomedicines-11-01068-f005:**
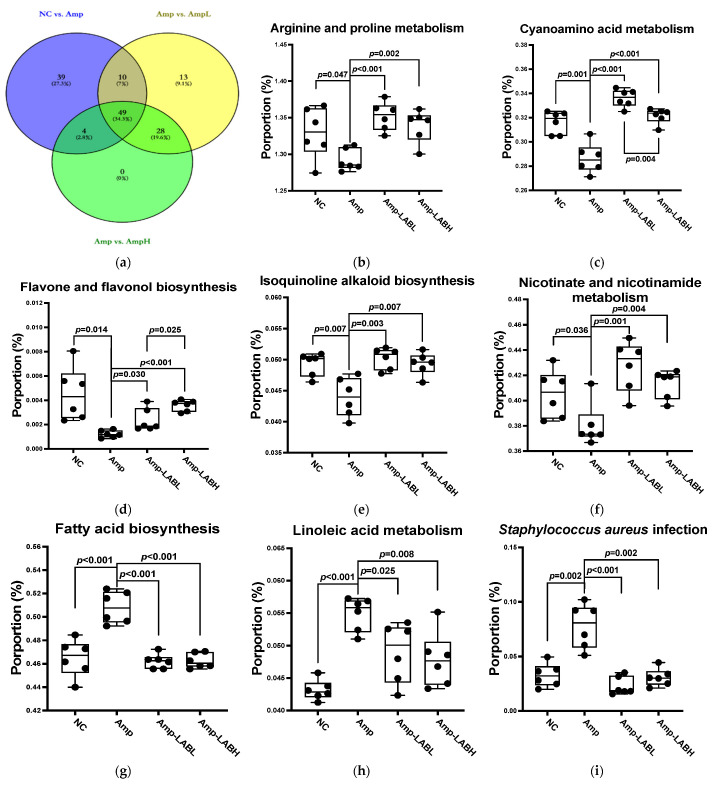
Effect of *Lactobacillus* spp. on gut microbes that regulate functional pathways in C57BL/6J mice. Based on the analysis of 16S ribosomal RNA V3-V4 gene expression in the ileum, PICRUSt2 software was used for functional pathway prediction. (**a**) Venn diagram analysis of different functional pathways between the four groups. The dramatic difference between the Amp and LAB groups includes 49 pathways, such as (**b**) arginine and proline metabolism, (**c**) cyanoamino acid metabolism, (**d**) flavone and flavonol biosynthesis, (**e**) isoquinoline alkaloid biosynthesis, (**f**) nicotinate and nicotinamide metabolism, (**g**) fatty acid biosynthesis, (**h**) linoleic acid metabolism, and (**i**) *Staphylococcus aureus* infection.

**Figure 6 biomedicines-11-01068-f006:**
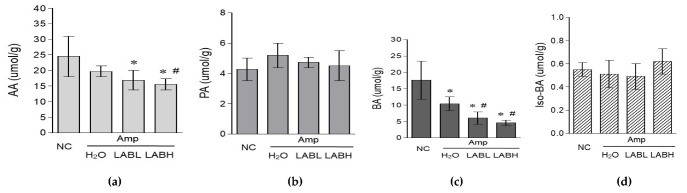
Effect of *Lactobacillus* spp. on short-chain fatty acid levels in C57BL/6J mice. C57BL/6J mice (*n* = 6/group) were grouped into the NC and Amp groups. The groups given the three-strain combination were fed a dose of 1.6 × 10^8^ CFU/mouse (LABL) or 4.8 × 10^8^ CFU/mouse (LABH) from the 3rd day until the 7th day. All groups were analyzed for short fatty acid levels in cecal content samples on the 12th day. Each SCFA standard compound was used for quantitative analysis. The levels of each SCFA were provided in units of “µmol/g cecum content”. Then, the levels were presented as “the percentage of individual SCFA/total SCFA.” (**a**) Relative levels of retinoic acid, (**b**) butyric acid, (**c**) acetic acid, and (**d**) isobutyric acid. *, compared with NC group, *p* < 0.05; #, compared with Amp group, *p* < 0.05.

**Figure 7 biomedicines-11-01068-f007:**
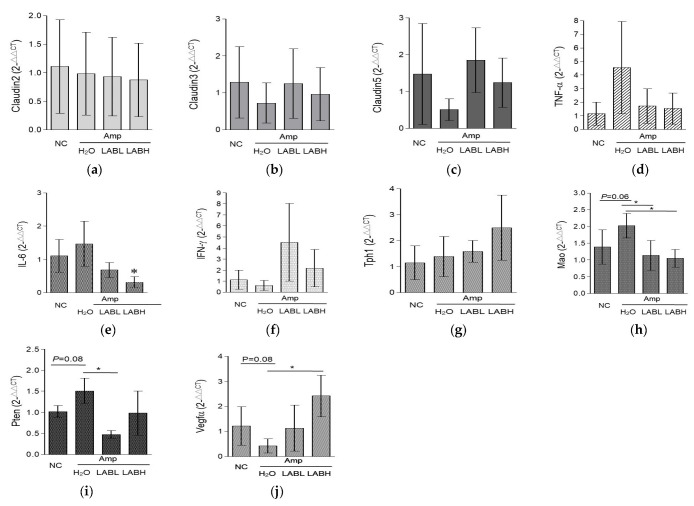
Effect of *Lactobacillus* spp. on mRNA levels of tight junction proteins and inflammatory factors in C57BL/6J mice. C57BL/6J mice (*n* = 4–5/group) were grouped into the normal control (NC) group and Amp group (Amp). The groups given the three-strain composite formula were fed a dose of 1.6 × 10^8^ CFU/mouse (low dosage, LABL) or 4.8 × 10^8^ CFU/mouse (high dosage, LABH) from the 3rd day until the 7th day. All groups were analyzed to determine the mRNA levels of tight junction and inflammatory factors in ileum content samples on the 12th day. qRT-PCR analysis of the expression of genes known to be tight junction proteins and inflammatory factors: (**a**) Claudin2, (**b**) Claudin3, (**c**) Claudin5, (**d**) IL-6, (**e**) TNF-α, (**f**) INF-γ, (**g**) Tph1, (**h**) Mao, (**i**) Pten, and (**j**) Vegfα levels. *, compared with the Amp group, *p* < 0.05.

**Table 1 biomedicines-11-01068-t001:** *Lactobacillus* species screened.

NO.	Species	Group	Rate of Recovery from Depressive Behaviors
			(%, Mean ± SEM)
1	*L. rhamnosus;* GMNL-74	A1 ^#^	107.4 ± 25.9
2	*L. reuteri*	A2 ^#^	45.2 ± 27.4
3	*L. fermentum*	A3 ^#^	77.4 ± 19.4
4	*L. plantarum;* GMNL-141	A4 ^#^	156.2 ± 46.7
5	*L. acidophilus;* GMNL-185	A5 ^#^	132.1 ± 54.0
6	*L. casei*	A6 ^#^	69.4 ± 14.0
7	*L. fermentum*	A7 ^#^	5.4 ± 22.1
8	*L. paracasei*	A8 ^#^	39.2 ± 15.3
9	*L. rhamnosus*	A9 ^#^	44.6 ± 14.0
10	*L. plantarum;* GMNL-662	A10 ^#^	10.8 ± 38.8
11	*L. rhamnosus* + *L. plantarum* + *L. acidophilus*	A1 + A4 + A5 ^#^	171.4 ± 71.8
12	*L. rhamnosus* + *L. plantarum* + *L. acidophilus*	A1 + A4 + A5 ^&^	234.3 ± 64.4

^#^: low dose: 1.6 × 10^8^ cfu/mouse (4 × 10^10^ cfu/60 kg person/day); ^&^: high dose: 4.8 × 10^8^ cfu/mouse (1.2 × 10^11^ cfu/60 kg person/day).

**Table 2 biomedicines-11-01068-t002:** List of primers used to analyze mice intestinal gene sequences used in RT-PCR analysis.

Gene	Primer Sequence (5′–3′) *	Size (bp)	GenBank Accession No.
GADPH (mice)	F-GCACAGTCAAGGCCGAGAAT	151	JN958248.1
	R-GAATCCTTCTGACCCATGCC		
Claudin2	F-TCCGGGACTTCTACTCACCA	190	XM_021188667.2
	R-CTCCTAGTGGCAAGAGGCTG		
Claudin3	F-GTTTCGGCATTCATCGGCA	180	XM_021186977.2
	R-TGCCAGTAGGATAGACACCAC		
Claudin5	F-GCTCTCAGAGTCCGTTGACC	235	NM_013805.4
	R-CTGCCCTTTCAGGTTAGCAG		
IL-6	F-TCTCTCCGCAAGAGACTTCCA	235	XM_021191538.1
	R-ATACTGGTCTGTTGTGGGTGG		
TNFα	F-CTAGCCAGGAGGGAGAACAG	149	NM_001278601.1
	R-GCTTTCTGTGCTCATGGTGT		
IFNγ	F-TCCTTTGGACCCTCTGACTT	130	NM_008337.4
	R-GTAACAGCCAGAAACAGCCA		
Tph1	F-ACTGCGACATCAGCCGAGAA	162	XM_036152912.1
	R-CGCAGAAGTCCAGGTCAGAAATC		
Mao	F-GGAGAAGCCCAGTATCACAGG	113	NM_173740.3
	R-GAACCAAGACATTAATTTTGTATTCTGAC		
VEGF-α	F-GCT ACT GCC GTC CGA TTG A	163	NM_001025257.3
	R-ATG GTG ATG TTG CTC TCT GA		
Pten	F-GGA AGT AAG GAC CAG AGA CAA	287	XM_006526769.3
	R-CAC CAC ACA CAG GCA ATG		

* F: Forward primer; R: reverse primer.

**Table 3 biomedicines-11-01068-t003:** Egg NOG functional annotations of LAB.

Pathway (Significance, *p* < 0.05)	Amp vs. NC	Amp-LABL vs. Amp	Amp-LABH vs. Amp
Amoebiasis	I *	D	D
Arginine and proline metabolism	D	I	I
Atrazine degradation	I	D	D
Bacterial toxins	D	I	I
Biosynthesis of siderophore group nonribosomal peptides	I	D	D
Biosynthesis of vancomycin group antibiotics	D	I	I
Cellular antigens	D	I	I
Chromosome	D	I	I
Cyanoamino acid metabolism	D	I	I
Cytoskeleton proteins	D	I	I
D-Alanine metabolism	I	D	D
D-Arginine and D-ornithine metabolism	I	D	D
Energy metabolism	D	I	I
Fatty acid biosynthesis	I	D	D
Flavone and flavonol biosynthesis	D	I	I
Glycerolipid metabolism	I	D	D
Glycerophospholipid metabolism	I	D	D
Isoquinoline alkaloid biosynthesis	D	I	I
Linoleic acid metabolism	I	D	D
Meiosis–yeast	I	D	D
Membrane and intracellular structural molecules	D	I	I
Metabolism of cofactors and vitamins	I	D	D
Nicotinate and nicotinamide metabolism	D	I	I
Nitrogen metabolism	I	D	D
Other ion-coupled transporters	I	D	D
Penicillin and cephalosporin biosynthesis	D	I	I
Phosphonate and phosphinate metabolism	I	D	D
Phosphotransferase system (PTS)	I	D	D
Plant-pathogen interaction	D	I	I
Polyketide sugar unit biosynthesis	D	I	I
Pores ion channels	D	I	I
Primary immunodeficiency	D	I	I
Protein folding and associated processing	I	D	D
Protein processing in the endoplasmic reticulum	D	I	I
Proximal tubule bicarbonate reclamation	D	I	I
Pyruvate metabolism	I	D	D
Renal cell carcinoma	I	D	D
Riboflavin metabolism	I	D	D
Shigellosis	I	D	D
Signal transduction mechanisms	I	D	D
Staphylococcus aureus infection	I	D	D
Streptomycin biosynthesis	D	I	I
Sulfur metabolism	I	D	D
Sulfur relay system	I	D	D
Taurine and hypotaurine metabolism	D	I	I
Tetracycline biosynthesis	I	D	D
Ubiquitin system	I	D	D
alpha-Linolenic acid metabolism	I	D	D
beta-Lactam resistance	D	I	I

* I: Increase; D: Decrease.

## Data Availability

The datasets generated and/or analyzed during the current study are available on the DDBJ BioProject listing page. https://ddbj.nig.ac.jp/resource/bioproject/PRJDB14147 accessed on 3 August 2022. The released project(s) will also be available on the NCBI BioProject listing page. http://www.ncbi.nlm.nih.gov/bioproject/ accessed on 3 August 2022.

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
