# Peer review of "Suppressive Effects of Lactobacillus on Depression through Regulating the Gut Microbiota and Metabolites in C57BL/6J Mice Induced by Ampicillin"

_biomedicines, 2023, doi:10.3390/biomedicines11041068_

Round 1

Reviewer 1 Report

The article is very interesting before, it could be considered for further publication I have some major quires which authors need to incorporate in the revised version ofthe manuscript for consideration of publication in the journal.

TitleThe title should be “Suppressive effects of Lactobacillus on depression through regulation of the Gut microbiota and metabolites in C57BL/6J mice induced by Ampicillin”

Abstract section does not give proper information.It is not clear what are the three anti-depressive potentials of LABs that undergoes combination to produce low and high LABs. The methodologyadoptedfor the study is not given. The introductory part needs to be rephrased. Activation of different metabolite pathways was studied, but to what context is not clear. Also, highlight essentialities and future perspectives of the study.

In the introductory section,

The authors need to add updated information by making use of the following papers:

Bhat MA, Mishra AK, Tantray JA, Alatawi HA, Saeed M, Rahman S, Jan AT. (2022) Gut microbiota and cardiovascular system: An intricate balance of health and the diseases state. Life. 12(12): 1986.

Paray BA, Albeshr MF, Jan AT, Rather IA. (2020)Leaky Gut and autoimmunity: An intricate balance of individuals health and the diseased state. Int. J. Mol. Sci. 21(24):9770

Khan S, Imran A, Malik A, Chaudhary AA, Rub A, Jan AT,Syed ZB, Rolfo C. (2018)Bacterial imbalance and gut pathologies: Association and contribution of E. coli in inflammatory bowel diseases. Crit Rev Clin Lab Sci. 56(1): 1-17

Lone JB, Koh WN, Parray HA, Paek WK, Lim JH, Rather I, Jan AT.(2018)Gut microbiome: Microflora association with obesity and obesity related commorbidities. Microb Pathogen. 124: 266-271

Methodology

The authors have performed 16S rRNA PCR and used different sequencing platforms. Have the authors submitted the results at any GenBank portal, if so, What are the accession numbers under which they are submitted?Which software was used for similarity search and phylogenetic studies.

Results

Figures: What does * and # represents is not clear. Please check it

Fig. 3a:Legends to figure is written in small font and are not clear. Please make them visible. Similar is the case with Fig. 5a.

Fig. 4:What interpretations were observed about finding members of different bacterial groups? What their presence indicates in the study needs to be highlighted.

Discussion:

The discussion section’s sentences are too long and the whole section is too lengthy. It is advised to make is crisp and clear. It is better to add references to literature of any suitable sources where used for correlation.

Conclusion

Please update your conclusion in light of recent reports from 2019-22. Do add a short future perspective of the study.

Author Response

Responses to the reviewer’s comments and suggestions are response point-to-point as follows.

Reviewer 1 

The article is very interesting before, it could be considered for further publication I have some major quires which authors need to incorporate in the revised version of the manuscript for consideration of publication in the journal.

  1. Title The title should be “Suppressive effects of Lactobacillus on depression through regulation of the Gut microbiota and metabolites in C57BL/6J mice induced by Ampicillin”

Thanks a lot for the suggestion.

The title was revised as "Suppressive effects of Lactobacillus on depression through regulation of the Gut microbiota and metabolites in C57BL/6J mice induced by Ampicillin".

  1. Abstract section does not give proper information. It is not clear what are the three anti-depressive potentials of LABs that undergoes combination to produce low and high LABs. The methodology adopted for the study is not given. The introductory part needs to be rephrased. Activation of different metabolite pathways was studied, but to what context is not clear. Also, highlight essentialities and future perspectives of the study.

Thanks a lot for the comment.

  1. The three anti-depressive potentials of LABs were added and described in the "abstract" section (in lines 21-22).
  2. The methodology was added in the "abstract" section (in lines 24-28).
  3. The introductory part in the "abstract" section was added, and re-write as "Depression is a medical and social problem. Multiple metabolites and neuroinflammation regulate it". (in lines 18-19)
  4. The activation condition of different metabolite pathways was added at line 28.

  1. In the introductory section,

The authors need to add updated information by making use of the following papers:

Bhat MA, Mishra AK, Tantray JA, Alatawi HA, Saeed M, Rahman S, Jan AT. (2022) Gut microbiota and cardiovascular system: An intricate balance of health and the diseases state. Life. 12(12): 1986.

Paray BA, Albeshr MF, Jan AT, Rather IA. (2020)Leaky Gut and autoimmunity: An intricate balance of individuals health and the diseased state. Int. J. Mol. Sci. 21(24):9770.

Khan S, Imran A, Malik A, Chaudhary AA, Rub A, Jan AT,Syed ZB, Rolfo C. (2018)Bacterial imbalance and gut pathologies: Association and contribution of E. coli in inflammatory bowel diseases. Crit Rev Clin Lab Sci. 56(1): 1-17

Lone JB, Koh WN, Parray HA, Paek WK, Lim JH, Rather I, Jan AT.(2018)Gut microbiome: Microflora association with obesity and obesity related commorbidities. Microb Pathogen. 124: 266-271.

Thanks a lot for the comment.

These references were added in the “Introduction” section as references in lines 43-47.

  1. Methodology

The authors have performed 16S rRNA PCR and used different sequencing platforms. Have the authors submitted the results at any GenBank portal, if so, What are the accession numbers under which they are submitted?Which software was used for similarity search and phylogenetic studies.

Thanks a lot for the comment.

The 16S rRNA PCR and bio information analysis results are stored at DDBJ Center and NIH bioproject. The above information is present in the “Data Availability Statement” section (in page 18, lines 647-650).

  1. Results

Figures: What does * and # represents is not clear. Please check it

Thanks a lot for the comment.

The * and # represents is statistical analysis results. It has a presence in each Table and Figure legend.

  1. Fig. 3a:Legends to figure is written in small font and are not clear. Please make them visible. Similar is the case with Fig. 5a.

Thanks a lot for the comment.

The legends in Figures 3a, 4a, 4b, and 5a have improved to make them visible. Please check Figures 3, 4, and 5.

  1. Fig. 4: What interpretations were observed about finding members of different bacterial groups? What their presence indicates in the study needs to be highlighted.

Thanks a lot for the comment.

This study to identifiably which gut microbiota was regulated by LABs. It will know the relationship between these microbiota compositions and depression-like behavior. This information was described in the “discussion” section (in page 16, lines 491-513 and 514-523).

  1. Discussion:

The discussion section’s sentences are too long and the whole section is too lengthy. It is advised to make is crisp and clear. It is better to add references to literature of any suitable sources where used for correlation.

Thanks a lot for the comment.

To shorten the "discussion" section, we re-write some of the sentences and paragraphs in the "Discussion" section. Including page 16, lines 495-513 and 531-538.

  1. Conclusion

Please update your conclusion in light of recent reports from 2019-22. Do add a short future perspective of the study.

Thanks a lot for the comment.

We have re-write the conclusion on page 23, lines 624-628.

We have added a future perspective in the “discussion” section (in pages 23, lines 745-751).

Reviewer 2 Report

The manuscript by Wan-Hua Tsai et al. investigated the suppressive effects of Lactobacillus on depression through regulating the gut microbiota and its metabolites. The study is of interes to the readers and I have the following suggestions and comments:

1, In the title, depressive should be depression. 

2, Moderate English changes are required for the manuscript. For example, in the abstract,  "there are three" should be "three",  "administered to C57BL/6 mice" should be " were administered to C57BL/6 mice". The authors must carefully check the English of the manuscript. 

3, Table 2 should be in a three-line formate. 

4, The authors must refine the panel A in figure 3. I can hardly read the lebelings. Same thing for the panels A and B in figure 4 and figure 5. 

5, Lefse LDA results in figure 4 should be added. 

Author Response

Responses to the reviewer’s comments and suggestions are response point-to-point as follows.

Reviewer 2

The manuscript by Wan-Hua Tsai et al. investigated the suppressive effects of Lactobacillus on depression through regulating the gut microbiota and its metabolites. The study is of interes to the readers and I have the following suggestions and comments:

1, In the title, depressive should be depression. 

 Thanks a lot for the suggestion.

The title was revised as "Suppressive effects of Lactobacillus on depression through regulation of the Gut microbiota and metabolites in C57BL/6J mice induced by Ampicillin".

2, Moderate English changes are required for the manuscript. For example, in the abstract, "there are three" should be "three",  "administered to C57BL/6 mice" should be " were administered to C57BL/6 mice". The authors must carefully check the English of the manuscript. 

Thanks a lot for the comment and reminder.

We already change the "there are three" into "three", and "administered to C57BL/6 mice" into "were administered to C57BL/6 mice" in the abstract section (in page 1, lines 20, and 23). And, we have carefully checked the English of the manuscript.

The original manuscript has been revised by an English professional editor has edited.

3, Table 2 should be in a three-line formate. 

Thanks a lot for the comment.

The table has been revised. (in page 6).

4, The authors must refine the panel A in figure 3. I can hardly read the lebelings. Same thing for the panels A and B in figure 4 and figure 5. 

Thanks a lot for the comment.

The legends in Figures 3a, 4a, 4b, and 5a have improved to make them visible. Please check Figures 3, 4, and 5.

5, Lefse LDA results in figure 4 should be added. 

Thanks a lot for the comment.

The Lefse LDA results are in figure 4c (in page 10).

Round 2

Reviewer 1 Report

The authors have incorporated my suggestions.. The Manuscript can be accepted in present form. 

Reviewer 2 Report

The authors have critically revised the manuscript. It can be considered for publication.